# Rethinking Family-Centred Design Approach Towards Creating Digital Products and Services

**DOI:** 10.3390/s19051232

**Published:** 2019-03-11

**Authors:** Jure Trilar, Veronika Zavratnik, Vid Čermelj, Barbara Hrast, Andrej Kos, Emilija Stojmenova Duh

**Affiliations:** 1ICT Department, Faculty of Electrical Engineering, University of Ljubljana, Tržaška c. 25, 1000-Ljubljana, Slovenia; Veronika.Zavratnik@ltfe.org (V.Z.); Vid.Cermelj@ltfe.org (V.Č.); Andrej.Kos@ltfe.org (A.K.); Emilija.Stojmenova@ltfe.org (E.S.D.); 2Faculty of Sport, University of Ljubljana, Gortanova 22, 1000-Ljubljana, Slovenia; Barbara.Hrast@fsp.uni-lj.si

**Keywords:** family-centred design, interdisciplinary approach, human developmental stages, qualitative research methods, sensors, smart city, smart community, smart family

## Abstract

This article provides further study of a family-centred design approach model established in previous studies, which aims to correspond to the limitations and needs of modern families using information and communication technology (ICT) solutions for common activities, communication and organisation of family time. The ambition is to systematically define and design features (functionalities) of a prototype solution that connects family members; provides proper communication; promotes active quality family time, active life, a health-friendly lifestyle and well-being; and uses various sensor- and user-based data sources through a smart city ecosystem platform. The original approach model was applied in designing the MyFamily progressive web application prototype solution as part of the EkoSmart: Active Living and Well-Being Project (RRP3) funded by the Republic of Slovenia and the European Regional Development Fund Investing in Your Future program. Extensive testing of the prototype solution used and the triangulation method used within thematic analysis for user interviews provide new insights and proposals for the change of the family-centred design approach model in the form of distinct developmental goals narrative for each generation to enhance motivation and relevance of content to different generations of users of such digital solutions.

## 1. Introduction

Digitization increases the importance of information, data and knowledge as crucial concepts of our everyday activities. The Internet of things and sensor ecosystems, services for big data and cloud infrastructure are emerging to support intelligent user-centered and social community systems [1]. For individuals, entangled in the interpersonal dynamics of various social structures, it is often difficult to adapt to evolution in the everyday use of information and communication technology (ICT). Individual’s rapid lifestyle pace often decreases the importance of family as an intimate, long-lasting community cell, and decreases the potential of intergenerational relationships within such groups. This could be demanding in terms of general well-being, health and a healthy long-term relationship. Given the distinct situation in the modern world, families are often burdened by the excessive use of technology by some members, usually the younger generation, and the underutilised use of ICT solutions by others, especially the elderly.

There is a lack of systematic integration of the family members’ distinct needs in strategies for designing digital services. To tackle communication between family members and other challenges in the family, we strive to develop an innovative ICT solution that will enable better interpersonal communication and organization of common family activities, including household work, in order to alleviate stress, ensure a more satisfactory time for socializing and promote an active and healthy lifestyle among family members of different generations.

The authors of this paper, employed at the Faculty of Electrical engineering and Faculty of Sport within the University of Ljubljana, had an opportunity to study possible implementations of proper design approaches within the framework of the EkoSmart programme [2], which involves several aspects of modern families’ involvement in the Smart City concept. Together with several Slovenian private and public entities, the Faculty is participating in the EkoSmart program—a programme financed by the Republic of Slovenia and the European Regional Development Fund Investing in Your Future programme. The EkoSmart programme consists of six different projects, which include various solutions integrated in a common smart urban ecosystem, and focus on three key areas of the city in the future: health, an active lifestyle and mobility.

One of these projects is the Active Living and Well-Being project (RRP3), in which we are ambitious to develop approaches and prototype solutions to support ICT in families in order to connect their members, communicate, promote quality family time, active life, a health-friendly way of life and well-being. In the RRP3 project, we recognized the need to achieve this goal with solutions for active life and family well-being as a model of a lasting and intimate community in the form of the MyFamily application.

Initially, we reference the EkoSmart: Active Life and Well-Being Project and existing prototype solution. Then, we present the fundamentals of a family-centred design (FCD) approach in creating digital services aligned with ambitions established in previous studies. The interdisciplinary triangulation methodology approach is explained in the subsequent chapter, following the exposition of study participants, measuring instruments and procedures of testing, and the interview process. A detailed analysis of the results of user experience testing and interviews structured via thematic analysis follows. We conclude with a discussion concerning the implications of these study findings on the FCD approach, and a proposal for changes in the key aspects of this approach model. Limitations of the study that need to be conquered in future research are disclosed at the end of the article.

### 1.1. EkoSmart: Active Life and Well-Being Project

The aim of the Active Living and Well-Being Project (RRP3), funded by the Republic of Slovenia and the European Regional Development Fund Investing in Your Future programme, is to establish new design approaches and digital service prototype solutions developed in the scope of families in smart cities [3]. These solutions would promote better connectedness among family members, communication and quality family time, an active and health-friendly lifestyle and well-being, and integrate sensor- and user-based data sources into a smart city ecosystem platform that enables connectedness between the plethora of smart city digital services. 

### 1.2. MyFamily Prototype

The purpose of this chapter is to highlight the prior work on the prototype solution while primarily focusing on user testing within closed laboratory environment that was essential for the production of the final prototype solution, which we used in study addressed in this article.

To design a prototype solution that would correspond with Ekosmart: Active Life and Well-Being Project goals, several stages of iterative development process were employed [3]: identifying motivational factors in the use of e-health solutions, determining the concept of a healthy and active family in a smart city, creation of personas, mental models construction, interviews and surveys regarding everyday ICT usage patterns, paper prototype testing. Numerous functionalities were mentioned and investigated in various phases of the project, including, among others: family tree, family budget, educational content in the fields of mental and physical health, physical activity measured through several sensors, notifications, help alerts, management family profiles, safe driving, collecting points and competition with other families, task list, microsocial network, family goals, location monitoring and sharing, event creation, common calendar, etc.

For a final implementation, we have selected a narrower set resulting from a systematic approach that captures the users’ wishes and experiences. The set of selected functionalities included: task management, notifications, calendar, common goals, physical activity, gamification through collecting points and family members’ connection list. These functionalities were implemented in a user interface that included sections: dashboard, calendar, tasks and common goals (Figure A1). On the dashboard, a list of family members with their current status and contacts is presented. The user can notice their upcoming and uncompleted tasks that were assigned to them and cumulative family activity histogram, which is only visible if family members use sensor devices to track their physical activity. Under the main panel, there is a panel with common family goals and a panel for suggesting common activities that can translate either to tasks or to goals. The second section is a calendar where tasks are presented in standard calendar format. The next section, tasks, comprises of task overview and filter panels, where users can edit, assign, delete and view their or other family members’ tasks that would typically entail household work, logistics tasks, fitness and recreation, cultural or other amusement events and meals. The last panel, goals, is and overview of common family goals that are comprised of several tasks, include all or most family members and have longer term of completion thus are more important than simple tasks. Users can also edit their settings and invite family members through the settings menu.

These features were then tested in interactive prototype solution for usability and other user experience aspects. Experiments were conducted in controlled laboratory environment and entailed: task-based problem solving within application with usage scenarios to measure task-completion speed and critical usage errors to validate selected functionalities and search for process improvements.

We conducted user experience and usability tests on 8 and 9 May 2018. Twelve participants responded to our call, among which, seven were men and five were women. Mean age was 37.41 years. All were employed and almost all, except one with high-school diploma, had a university degree. The tests were conducted on a dedicated personal computer in designated room, where two moderators and one test subject were present at a given time. We measured completion times of 9 selected tasks. Task completion rate was 100%. Please see Table A1 for results regarding average task completion times.

Standard questionnaires System Usability Scale (SUS), User Experience Questionnaire (UEQ) and NASA-Task Load Index were additionally used in usability and user experience testing and helped us validate the chosen functionalities for interactive prototype. The System Usability Scale result was 82.30. Please see Figure A2 for User Experience Questionnaire results and Figure A3 for NASA-Task Load Index results.

These past testing activities were fit for use in controlled environment experiments and were the basis for further development of the MyFamily application prototype that we needed to study within the groups of end users in their homes for additional insights regarding real-life usage and overall FCD approach improvement, which is the focus of this article.

### 1.3. Family-Centred Design

In a previously published article entitled “ICT to Promote Well-Being within Families” [3] by our research team, we tried to establish a notion of a new approach towards systematically designing family-centred (digital) services. FCD is concerned with important life events, goals and constraints within families or other intimate long-lasting communities that need to be addressed in a (digital service) design process to create meaningful use case scenarios. We have established a few fundamental guides when dealing with designing services in the context of family, through the iterative research of families’ behaviours, ICT usage patterns, processes within and communication between family members. It was our assumption that this approach should entail:Analysis of common modes of communication,Assessment of access to technology among group members,Shared purpose in everyday tasks, andConcern that the processes in the group should be inclusive as much as possible in order to create a better experience for all.

In the previous article, where we described the development of the EkoSmart prototype (Figure A1) and the FCD approach, we proposed several distinct prototype solution functionalities based on the results of an iterative design and testing process as well as study analysis. On the basis of data analysis regarding device type usage and the FCD approach towards increased inclusion, we outlined a feature that would enable the inclusion of users in the form of different user interface modalities for different devices. The digitally unskilled elderly would be able to participate in family communication and organisational processes by using familiar technology in this instance, text messaging (SMS) on their regular mobile phones, whereas younger family members would use an enhanced interface on their smartphones or personal computers [3]. 

In our previous study, we highlighted positive attitudes towards the use of ICT in everyday life and that many participants see the most important role of ICT as a tool for planning and time management; even more so in the family usage context, where one of the most common reasons for not spending quality family time together was schedule planning inconsistencies. In congruence with these findings, we concluded that the basic functionality of the MyFamily prototype solution should involve task management used for logistics and household work coordination to promote communication and collaboration among family members. To encourage quality family time, it should also introduce potential recommendations for shared leisure activities [3].

To motivate the usage of an activity tracker and higher completion of tasks, which could indicate more solidarity in the family, we propose some extent of gamification mechanisms to be integrated in the prototype solution [3]. It was presented in the form of collecting experience points for activity and participation in common family activities. We assume that experience points are a concept familiar enough to all family members, the elderly, young and middle generation.

With the current study presented in this article, we aim to challenge and review some of these essentials. Testing user experience and conducting interviews with the users of a prototype solution designed with aforementioned principles provides us with enough insight and interpretive value to re-examine different assumptions, and to potentially extend this approach with additional, new concepts.

## 2. Method

In this chapter, we present the interdisciplinary methodology approach, comment on participants, and present the procedure and instruments used for the main study.

### 2.1. Methodology

In regard to the positivist (or postpositivist) paradigm, we would like to address requirements validation through an interdisciplinary interpretive approach that would provide adequate external validity through a logical-deductive premise to conclude whether this kind of qualitative insight can offer benefits to distinct design approaches towards creating new (digital) services.

As blurring of disciplinary boundaries, in the milieu of scientific and commercial research, is becoming more and more evident, calls for mixed methodology are now common. A (mono)disciplinary approach can be limited to capture challenges of modern (multidimensional and complex) phenomena. Scientific disciplines (e.g., ecology, chemistry, biology, psychology, sociology, economy, philosophy, linguistics, etc.) are categorized into distinct scientific cultures: the natural sciences, the social sciences and the humanities. Interdisciplinary research may involve different disciplines within a single scientific culture [4].

Interdisciplinary research design starts with the “conceptual design” which addresses the ‘why’ and ‘what’ of a research project at a conceptual level to ascertain the common goals pivotal to interdisciplinary collaboration [5]—then moving to a paradigm or perspective and after that to interpretive practices (or methods) that represent the empirical world [6].

Diverse definitions and labels try to establish this growing research position: multistrategy, multimethods, mixed methodology or mixed methods [7]. Use of triangulation in research has been the subject of different considerations. Some explain it as a contribution to a deeper understanding of the study phenomenon, others try to establish an argument that triangulation is one of the validity measures to increase the study accuracy [8]. In essence, triangulation is the use of at least two methodological approaches, theoretical perspectives, data sources, researchers or analysis methods in studying the same phenomenon in order to increase study credibility.

Throughout literature research, we encountered several types of triangulation: methodological triangulation, investigator triangulation, theoretical triangulation, analysis triangulation and data triangulation [8]. To narrow the choice of the theoretical basis according to our capabilities, in terms of the method and variety of data used, our estimate is that our study can benefit from focus on the theoretical and investigator triangulation approach.

Theoretical triangulation is defined as the use of multiple theories in the same study for the purpose of strengthening or refuting findings through a variety of examining same-dimensional factors from different theoretical perspectives [8]. That approach alone implies the involvement of experts from various fields in order to assure a broad and thorough enough overview of the research. Therefore, investigator triangulation will also be employed in this study. Investigator triangulation can be employed in a variety of ways, for example, interviews in a case study research can be conducted by employing more than one interviewer [9]. Investigator triangulation involves the use of multiple observers, interviewers and data analysts in the same study for better completeness purposes.

Unlike triangulation for confirmation purposes, with classical benefit for validation in the form of quantitative results by qualitative studies, the use of triangulation for completeness seems a better fit for interpreting findings of this study as it is mainly used in researching less explored or complex research problems [7]. One of the advantages of the qualitative research paradigm is generating a rich amount of data that further the range of concept investigation in order to help researchers developing hypotheses for future quantitative studies. After a large amount of data has been generated by a qualitative research method, a researcher has to employ quantitative research methods in the form of data collection methods and analysis to get a deeper and more comprehensive picture of the phenomenon under investigation [7].

In terms of the requirements of the engineering process, this would be a stage of requirements validation—checking that the documented requirements and models are consistent and meet the needs of the stakeholder.

Our conceptual design entails focus on important life events, goals and constraints in the developmental stages of an individual who functions in a network of long-lasting meaningful relationships that have extensive impact on their behaviour patterns in order to extract some valuable information regarding the conception of functional features used in meaningful digital services use case scenarios.

In regard to the technical design of the research, input data will not be operationalised in quantitative measures; rather, we will use a “soft”, qualitative approach. Semistructured interviews on the topic of users’ experience with an advanced web application that tries to address particular needs of a modern family will be held after a certain testing time frame. After collecting and recording interviews with participants in accordance with the designated pluralistic approach, a team of experts from different fields will separately review the content and make remarks that can be interpreted with their distinct expertise, although with an overall focus on important life events, goals and traits of the participants.

To analyse these interviews, we employ thematic analysis—a structured approach to discover themes and concepts embedded throughout interviews. Thematic analysis is a method for identifying, analysing, and reporting patterns (themes) within data [10]. The thematic analytic method and procedure is thoroughly described by Braun & Clark [10], and their article serves us as a manual on how to address the interview analysis in a flexible yet structured manner.

Our vision is that knowledge from the fields: developmental psychology, anthropology, sociology, user experience design and computer application programming will provide us with enough coverage to detect possible valuable themes regarding features (users’ needs) that were not (properly) addressed in previous iterative stages of the EkoSmart prototype development.

A more detailed process of our inquiry is described in adjacent chapters “Participants”, ”Instruments” and “Procedure”.

### 2.2. Participants

Initially, 8 families with more than 30 individuals, family members, responded to our call for testing the MyFamily interactive prototype for a period of approximately two weeks. Five families with 20 participants were genuinely active during the testing period, and they took part in the initial and subsequent user experience surveys and semistructured interviews that we included in our data sets.

Four of the active families were from urban or suburban areas, and one family was from a predominantly rural environment. One member did not have any in-person contact with their family during the testing period.

Four members of families were children (up to 18 years old), 12 were adults (18 to 65 years old), and 4 were seniors (more than 65 years old). Ten men and ten women were included in testing and interviewing. All adults were employed or self-employed. All seniors were in retirement. The mean age of the participants was 40.65 years.

### 2.3. Procedure

Semistructured interviews were conducted after a two-week period using the MyFamily progressive web application within the family. Five interviews with families were recorded between August and October 2018. Interviewers asked the participants a series of open-ended questions regarding the ease and frequency of use, technical and usability problems they may have encountered, their behaviour regarding the use of ICT in family communication and organisation as well as visual and overall experience of the app. Interviews lasted from 35 min to 1 h and 15 min. Interviews with families were transcribed verbatim. Participants were also asked to keep a diary of subjective remarks regarding usage during the testing period. Those diaries were analysed in combination with user interviews.

In addition, standard usability testing forms were distributed among participants to acquire an insight into the perceived usability and user experience of the prototype before and after the usage. These testing forms included: a User Experience Questionnaire (UEQ) to measure 6 dimensions of user experience with the product, Standard Usability Scale (SUS) for subjective assessments of usability, and the NASA-TLX questionnaire for cognitive workload measurements. These questionnaires were filled in before and after actual prototype testing in order to validate and assess previous efforts in the iterative design approach of an interactive prototype.

Our main goal was to improve and broaden the understanding of family needs and attitudes towards the specific usage of certain, also nonimplemented, features. This would be conducted outside of the user experience and prototype usability testing scope with an analysis of themes that were most common among different families. The thematic analysis procedure [10] was chosen as the analytic method. In our analysis, we followed the stages of thematic analysis as described by Smahelova et al. [11]:Familiarising yourself with your data and transcription of verbal data.Generating initial codes and grouping them into topics.Searching for themes.Reviewing themes.Defining and naming themes.Producing a report.

The corpus of our data comprised all interview transcriptions and user diary entries. Sets that were used consisted of clear and meaningful answers from participants. We generated these sets with the triangulation method approach—every researcher created their own set via listening and reading transcripts with making remarks for relevant topics (Figure A4). Designated codes created upon initial readings of sets and discovering related topics were grouped into themes. Themes that would emerge from these codes were then grouped into three main categories for additional clarity: confirmation of functionalities, problems and criticism of functionalities, and suggestions for improvement. That enabled us to extract meaningful interpretations of the results regarding the prototype user experience, user needs, behaviour patterns and expectations towards this kind of digital services and products.

### 2.4. Instruments

Standardized questionnaires measuring subjective user experience, system usability and task load are used alongside user interviews to gain deeper understanding of the prototype solution usage in the testing period.

#### 2.4.1. User Testing

Standardized subjective questionnaires provided us with comparable results between families. We were able to correlate our results with the results from other research studies in the same field. To measure usability of the application, the System Usability Scale [12] was used. User experience was measured using the User Experience Questionnaire [13]. Subjective workload was measured using the NASA Task Load Index [14].

Users were asked to fill all the questionnaires immediately after the first user testing without talking to other members. At the end of two weeks, they were asked to retake the questionnaires again with all the knowledge and experience they gained in the testing period.

##### System Usability Scale

The questionnaire System Usability Scale (SUS) is a tool that provides a “quick and dirty” scale for measuring usability of an application. SUS is widely used and has become an industry standard. It is a Likert scale that consists of 10 statements. The user has to choose one option for each statement; from Strongly disagree to Strongly agree. SUS measures learnability (items 4 and 10), and usability (other items) [15]. Usability covers effectiveness (the ability of users to complete tasks using the system), efficiency (the level of resources consumed in performing tasks), and satisfaction (users’ subjective reactions to using the system) [12]. Results from the SUS questionnaire were compared to results gathered from more than 500 different studies. The percentile curve used to interpret the results is presented in Figure A5.

##### User Experience Questionnaire

The User Experience Questionnaire (UEQ) contains 6 scales with 26 items in total. They measure Attractiveness (Overall impression of the product), Perspicuity (Is it easy to get familiar with the product?), Efficiency (Can users solve their tasks without unnecessary effort?), Dependability (Does user feel in control of the interaction?), Stimulation (Is it exciting and motivating to use the product?) and Novelty (Is the product innovative and creative?) [16].

Attractiveness has 6 items, other scales have 4 items. Attractiveness is a valence dimension. Perspicuity, Efficiency and Dependability are pragmatic quality aspects, which means they are goal-directed, while Stimulation and Novelty are hedonic quality aspects and are not goal-directed [16].

##### NASA-Task Load Index

The NASA Task Load Index (NASA-TLX) is a tool that measures subjective workload assessment. It is made up of six factors: Mental Demand (How much mental and perceptual activity was required?), Physical Demand (How much physical activity was required?), Temporal demand (How much pressure did you feel due to the rate or pace at which the task or task elements occurred?), Own Performance (How successful do you think you were in accomplishing the goals of the task?), Effort (How hard did you have to work?) and Frustration (How insecure, discouraged, irritated, stressed and annoyed were you?). Three dimensions relate to demands imposed on the subject (Mental, Physical and Temporal), and three to the interaction with the task (Effort, Frustration, Performance). Each factor has its own scale that goes from 0 to 100 [14].

#### 2.4.2. User Interviews

Contrary to the standard usability tests that were conducted at the beginning and the end of the testing period, we made qualitative interviews only after the testing was completed. In line with the principles of triangulation, we used different methods and approaches to analyse and interpret the interviews, and to validate our measures [7]. With the use of triangulation, we also enabled a broad context for further qualitative analysis. 

As previously explained, each of the researchers from different fields of work (sociology, anthropology, psychology, and computer science) separately analysed user diaries and interviews and made some personal remarks. Stemming from these remarks, we agreed upon creating three main topics to cluster the data in such a way that we could elaborate further (see Figure A4) through: (i) Confirmation of functionalities, (ii) Problems and Criticism of Functionalities, and (iii) Suggestions for improvement. To make our topics more precise, we defined themes for each of them. These helped us to make meaningful connections between the data obtained with qualitative methods and the data obtained with standardized usability tests and monitoring of users’ activities in the prototype solution interface. Our further comments on the results of user experiences in connection with the developmental stages of the participants are also based on those codes.

## 3. Results

In this chapter, we display the results acquired from the participants during the testing period with the use of selected methodological instruments.

### 3.1. User Testing

Standardized questionnaires measuring system usability, subjective user experience and task load were used before the testing period started and after the interviews very conducted at the end of user testing period.

#### 3.1.1. System Usability Scale

Participants’ answers varied between age groups. This prompted us to analyse the results by age groups and not by the sum of all means (Table 1). Four participants were older than 65 years; four were younger than 18 years and 12 were in between. The results are interpreted using the adjective scale [17] and percentile rank (Figure A5) [18].

Youth did not have any considerable problems learning the application and stated that they would not need an assistant to help them. Their mean SUS score at the beginning of the testing was okay (41st percentile). Results at the end of user testing were excellent and ranked in the 90th percentile. This might be due to the experience young people have with mobile and computer applications. Two weeks were enough for them to learn and use the application.

Adults had more problems and thought that they should learn many things before independent use of the application. This was not as big of a problem as they stated because most of them learned how to use the application by trial and error. Score at the beginning was okay and ranked in the 38th percentile. Results at the end were slightly better, but the score was still okay (ranking in the 42nd percentile). Few users changed their opinion (users with a higher SUS score at the beginning had lower at the end and vice versa). 

Seniors thought that they would have problems learning the application. This was shown in the score after the first use and even more in the SUS questionnaire at the end of testing. The mean of the SUS score at the beginning was okay (29th percentile). The score was slightly lower at the end of user testing, ranking it in the 23rd percentile. Results in percentile this low mean that the usability of the application is not appropriate for seniors. Enhancing usability for seniors should be our top priority.

The mean SUS score for all users at the beginning and the end of user testing was below average. The score of youth and adults was higher at the end of user testing. We were able to get a lot of information about problems that lowered the usability of the application through user interviews. Many problems arose from the inadequately presented application functionality. Users were not able to properly learn how to use the application and this led to lower overall usability (effectiveness, efficiency, satisfaction).

#### 3.1.2. User Experience Questionnaire

There were not enough families included in user testing, so we were not able to conduct an extensive quantity analysis. We were able to track trends that complemented data collected from user interviews. Comparison between questionnaire data gathered after the first interaction and data collected after the last interaction is shown in Figure 1. Results gathered after the first use was mostly positive and gave us affirmation that our idea solved the problems that the users had. Data gathered two weeks later, at the end of testing with real 3-generational families, presented worse results. The most worrying fall was in Stimulation. User input from interviews provided insightful information on different reasons why users were demotivated. 

Results can also be interpreted by attractiveness as well as hedonic and pragmatic quality. The mean of attractiveness was lower at the end of user testing. Users were excited after the first contact with the application, but these positive emotions wore off in two weeks of user testing. Task-related quality aspects were better rated at the end of testing. Results from categories in pragmatic quality were slightly above the UEQ benchmark. This tells us that the participants did not have major problems using the application. Hedonic quality aspects were poorly rated. This means that the participants had low motivation and the application did not convince them enough for further use. They were also not keen on the application design as it did not stand out to most of them.

More than one quarter of all users had at least one contradictory answer at the beginning of user testing, which might be due to their inexperience with the UEQ questionnaire. Results at the end had even more contradictory answers. After analysing suspicious answers, we decided not to include data from two participants, one in the beginning results and one in the end.

#### 3.1.3. NASA-Task Load Index

An unweighted NASA-TLX questionnaire was used at the beginning and at the end of user testing. Results between different users were inconsistent since they had a very different understanding of the scale (Figure 2). Most of the users thought that the use of the application was mentally demanding. Two users stated that the application was physically demanding. Temporal demand was also present but was not as high as mental demand (Figure 2).

Own performance and frustration had a strong correlation. Users that were not satisfied with their performance evaluated their frustration relatively similarly. Most of the users were pleased with their own performance. Few participants were very unmotivated, unsure or frustrated using the application and ranked their Frustration accordingly. Effort was comparable to mental demand with some deviation between participants.

Participants also solved this questionnaire at the end of user testing. Only results in mental demand were lower in comparison to the results at the beginning. Mean results in every other category were higher, with the largest change being frustration.

### 3.2. User Interviews

User interviews, conducted after prototype application testing period, were categorised into general three topics that provided analytical separation of different themes.

#### 3.2.1. Confirmation of Functionalities

We divided the category Confirmation of Functionalities in subcategories to gain an insight into two important fields: firstly, the confirmation of the results of our previous research work that was the basis of the development of the MyFamily application, and secondly, the confirmation of some application functionalities. Two most confirmed functionalities were the Calendar, which was considered a very practical collaborative calendar for the organisation of family life, especially for families with younger children, and Goals and Tasks. While Tasks were used only to organise the time in the near future, the main advantage of the Goals was that they were designed as long-term “obligations” and were therefore very good conceptual motivators and many times taken very seriously. The third confirmation was clustered in the sub-category Interpersonal Communication. Through establishing common Family Goals, communication between the family members was also encouraged. 

Especially for grandparents, connection to younger generations was indeed very important. But, one of the mothers told us that “actually, because of the application, you talk more often to each other; because of the tasks, because of collecting the points.” Equal rights in editing and changing common tasks were seen as a sign of mutual trust within the family and were therefore perceived as a good factor. The fourth confirmation factor was Gamification of tasks. Especially for the older generation of the participants, collecting points was a good motivational factor, for the other family members points were interesting but did not increase motivation considerably. Instead, for some users, the “feeling when you mark the task as completed” was the motivator in itself. The last but very important category was named General Impression. In general, the application was perceived as very transparent, “cute”, easy to use and holistic. None of the users reported major difficulties in using the functionalities. 

#### 3.2.2. Problems and Criticism of Functionalities

We clustered user problems and criticism into six subcategories connected to different aspects and functionalities of the application.

The dimension of *gamification/collection of points* received some criticism concerning a too wide flexibility in choosing how many points a specific task will be worth when completed. Moreover, to collect points and compete within the same family was mostly considered unnecessary; only two grandmothers reported that this was a motivational factor for them, all the other users suggested less competitive ways of gamification. It was also noted that collecting points was not “fun” and that “points make no sense”. Another remark was made in connection with the transparency of the points collected as it was unclear where one could find the number of points collected by other members of the family. Another functionality was perceived as unclear: *notifications*. Firstly, many of the users did not find the option to turn on their preferred notification type and therefore did not receive any reminders or notifications. The users wanted more stimulation and for the application to be more present and let them know what to do, where, when and with whom. Secondly, the notifications were very unstimulating and even though users received them, there was no real motivation in terms of interesting content suggestions for the tasks and goals. Thirdly, some users also had some problems with sending family members an invitation to join their family.

The next subcategory is connected to *accessibility*, especially in connection with the oldest generation of users. Besides very basic accessibility problems, such as the lack of contrast or too small icons, the main problem was that they were not familiar with the concept of mobile/web applications. Therefore, they were also not familiar with some specific functionalities or had problems with the registration and creation of tasks/goals. In general, the oldest generation expected more specific instructions as to how to use the application; for them, the application was mostly perceived as an “extra effort”. 

One of the most commonly mentioned criticism of the application was the *lack of possibilities for personalization*. There was a lack of possibilities to directly communicate with family members, and there were also too few avatars available if the family was big. Also, the application graphic was seen as not interesting/stimulating. In spite of the lack of personalization opportunities, in some cases, the application was paralleled with applications used at work. Further, *user problems* is a very broad subcategory encompassing a wide range of difficulties and criticism. Most of them are related to some specific functionalities, such as poor visibility of the Help button, the mode of setting the Date and Time was considered very impractical, Search Engine was non-transparent and so was Family Tree. The visibility of the Status of the family members was poor, there were difficulties in joining the Family or problems in the Registration process. One of the criticisms was connected to the functionality Tasks: it was not possible to copy the same task or set it as a recurring task. A very important criticism was also made in connection with not understanding the hierarchy Task vs. Goal. 

Lastly, but very importantly, we summarized some important insights in the subcategory *usefulness*, where we reflect upon the general perception of the usefulness of the application. The leading comment was that it was necessary to simplify the application and put a broader context in the forefront since very often it was not very well understood why the points were being collected. It was also commented that there were not many functionalities that support or can be integrated into the life of the family. The general usefulness of the application is questioned and the potentials of an improved version are discussed.

#### 3.2.3. Suggestions for Improvement

The last category was formed for two reasons: firstly, to comment upon the exposed criticism and shortcomings of the application in the context of suggested solutions, and secondly, to draw from suggested examples of other applications that the test users had experience with. We divided this category into six subcategories corresponding to the criticism and problems noted during the testing period. 

Gamification and the collection of points were suggested to be optional—the possibility to opt out should be considered. Evaluating tasks and goals with points should be limited, for example on a scale from 1 to 10 or 1 when the task is done and 0 when it is not completed. Also, more fun visual means should be included to make the application more entertaining. A possibility to compare results between different families would be a great asset as well as some additional awards when a task is completed or a goal achieved—especially in terms of badges or animations added to the name (for example an Apple, when someone was the first to complete the goal in the category of exercise etc.). The next cluster of suggestions was grouped in the Notifications subcategory. The main problem with notifications was that users did not turn them on, so the suggestion is to create an “opt out” functionality rather than “opt in”. In general, the application should be more present, especially in the form of notifications, reminders, motivational messages, warnings on the coming deadlines etc. One grandmother, for example, suggested that “if I have no task remaining, the application should say: ‘Congratulations, you have no tasks, you completed all of them’.” Because users use multiple applications, integration of notifications from other applications was also suggested. Settings for the notifications should also be more flexible in terms of setting the time when notifications will be sent out. Preprepared content in the sense of some already integrated functionalities turned out to be an important part of the application. Different contents related to the lifestyles of the users could be included, like Recipes, Family Albums, or some suggestions for specific tasks (like vacuuming, washing the dishes etc.). It is necessary to include more instructions for use; these could be included in each section promptly or as a general introduction to the application at the beginning, but in any case, the Help button should be more visible. An interesting suggestion was also connected with the Calendar that could automatically generate available dates and terms for different combinations of family members. 

The next two subcategories are closely connected: Personalization and Community Building. The former relates to making the application feel more personal, and the latter concerns making the application feel more family friendly. Suggestions on Personalization are explicitly connected with the criticism/problem encountered before: the colour scheme could be optional, avatars could be selective with the possibility to use photographs, the change of the name of the family should be made available, there should be more commenting options on different tasks. A platform for posting and exchanging photos should also be integrated. In terms of Community Building, it was suggested that a platform for exchanging family photos and creating family photo books should be integrated. The users also suggested redesigning the family tree to be more transparent, and also broadening its scope to include partners and other, more distant relatives (i.e., uncles, aunts, cousins etc.). The sense of community would be enhanced if means for family conversations—like messenger or family wall—were provided. This would also enable reminders to other family members, as was also suggested.

Important and insightful suggestions were made in connection with Tasks and Goals. It is very important to include the option of repetitive tasks and the possibility to copy the same task to different dates. Photographs could also be included when defining tasks and goals to illustrate the aims; in this sense, it would be very illustrative if comments could be added as well. When more family members are included and have to co-operate to complete a task/goal, it should be enabled that the task/goal can be only partially completed when a certain member finishes their part. In this case, it would be very illustrative if a progress bar was used instead of points. Because family goals are plans for long-term periods, it was also suggested that complementation of tasks when needed/changed should be enabled. The arrangement of tasks and goals could be made more transparent and easier to browse. In terms of more pragmatic functionalities, it should also be made more clear which fields are required when defining new tasks/goals—especially the oldest generation of users had problems understanding what was required and what optional. Lastly but importantly, it is necessary to make the process of creating the tasks and goals more available and faster—for example via a “fast button”. 

## 4. Discussion

Discourse about implications, deriving from the result of our methodology design, for existing FCD approach model is further extended with proposals for future development while also considering given limitations.

### 4.1. Implications of Results

Corresponding to the methodology procedure results, there were some important conclusions regarding the prototype solution as well as FCD approach we tried to establish in previous work. In regard to confirmations of functionalities, the overall impression was satisfactory for most of the testing participants; however, important improvements arise from exposition of criticism and suggestions for enhancement. 

Participants primarily reported problems about user experience concerning the lack of advanced personalisation and content sharing tools with more appropriate gamification mechanisms in the form of visual stimulation (badges) instead of collecting points. These relatively unproblematic user experience problems can be solved through established design and development approaches. 

We determined that most of the family members actively using the prototype solution used it on mobile phones, the most personal and “always-on” of devices involved in the study. In this view, we propose focus exclusively on mobile smartphone devices rather than different modalities on different devices. In web application development terms, this would be described as mobile-first approach.

As we can conclude from the study results, device type ownership is not a significant factor when it comes to inclusion via adapted modalities (desktop, smart phone, regular phone interfaces) for every stakeholder. Part of the population, predominantly elderly, did not understand the connectedness of a wholesome, originally thought, inclusive process if they only received degraded interaction via text messaging alerts regarding tasks sent to them from a fully-fledged web interface. This fundamental understanding of the purpose of the solution prototype was nonproblematic for the same demographic users using a fully-fledged interface on a smart mobile phone with all functionalities included.

On average, according to System Usability Scale questionnaires (Table 1), usability was sufficient among all participants. However, when we analyse results by demographic groups (young, adults, elderly), we can realize that although usability was rated good among the young and adult population, the elderly were extremely dissatisfied. When analysing interview remarks, we can also notice distinct problems concerning the elderly and the content of the prototype solution. The younger generation and adults understand content creation, whether it is task creation, information sharing, social media substance. The elderly, in particular, have almost no recognisable conception about that area of use. In our interviews, they often commented that the prototype solution is ‘empty’ and ‘has nothing to offer’, which implies that there was no preprepared content that would clarify the context of use or otherwise motivate them.

Through our investigation, it became apparent that the use of ICT could help alleviate an increasing problem of the aging population, loneliness. In our study, it is apparent that the elderly are the most vulnerable demographic group in regard to enhanced inclusion in the FCD approach. A number of elderly participants mentioned the importance of connectedness with their grandchildren and children. We believe that the opportunity to be in contact with other family members could be the most important motivational aspect and highly useful functionality for the elderly. There have been some studies that focused on the beneficial use of ICT in the elderly [19,20,21]. However, further research is needed to fully understand the effect that ICT could have on loneliness in the elderly and, subsequently, what interventions could be used in order to alleviate loneliness. 

In congruence with these findings, we have removed a fundamental component—type of inclusion provided via different modalities—from the FCD approach. This very important decision speculates that inclusion in common family activities should be based on a common narrative instead of device type ownership and perceived user competence. We believe that addition of narratives based on developmental life goals would benefit the overall model of the FCD approach more.

### 4.2. Proposal for Further Development of a Family-Centred Design Approach

Based on the results we have obtained and their implications, we argue that the addition of psychological developmental aspects of users may contribute to a better model for the development of the FCD approach. This could be an additional dimension that would help with finding functionalities that would prove to be useful and meaningful to the majority of users. 

We believe that life-span developmental psychology and, more specifically, a developmental tasks narrative could help us build a better model that would provide developers with tools of finding meaningful and useful functionalities for users of all ages. This is especially important in the FCD approach, where the conceptualization of a mobile or web application entails users of all ages. Life-span developmental psychology argues that human development takes place through all our life, from birth to death, and it studies the change and stability in human behaviour [22]. Individual’s behaviour is judged (by self and others) based on criteria that reflect expectations of age-appropriate and adaptive behaviour [23], and are grounded in normative developmental and contextual changes over individual’s life time. These behavioural criteria are called developmental tasks [24].

Robert Havighurst was the first to formally define the term developmental task as a combination between an individual need and a societal demand [25]. He proposed that a number of tasks need to be achieved over the life time, where each task is expected to be engaged in and mastered in a developmentally appropriate age period [24]. Havighurst [25] distinguished between tasks that stem from biological maturational processes, sociocultural pressures and those that arise from individual’s values and goals. He also emphasized that mastery and successful resolution of these tasks result in happiness and success in later tasks, and could therefore be regarded as motivational. 

For the purpose of this research and possible adaptation, we decided that three developmental stages (late childhood and adolescence, adulthood, late adulthood), and tasks associated with them could be enough to construct a working model based on the demographics of users in our interviews. Some of the widely recognized tasks in late childhood and adolescence are learning to read and write, getting along with peers, making friends, academic achievement, adjusting to pubertal changes, close friendships, romantic relationships, identity exploration. Developmental tasks in adulthood include establishing work, financial security, a committed romantic relationship or marriage, caring for dependent children, maintaining a household, maintaining satisfactory work, romantic relationships, launching children, caring for aging parents, adaptation to physical changes. In late adulthood individuals need to adjust to changes of aging, dependence on others, death of a spouse or close friends, transitioning to retirement, settling affairs [24]. It is important to note that some developmental tasks are considered universal, whereas some are culturally and contextually specific; they also emerge and change with changes in the ways of living and culture [24]. Therefore, our model roughly delineates between developmental stages and allows developmental tasks to be mastered whenever an individual feels ready for it. 

Changes in fundamentals of the FCD approach can be utilised in working functionalities to provide better motivation and with that more inclusiveness for all users involved in common activities over ICT. This would be achieved with preprepared content tailored to particular user’s age and developmental stage. It could be implemented in the form of suggestions for activities on top of existing user scenarios, such as adding a new task where there are also recommendations to whom it would be suitable to share. This would be a less intrusive introduction of a new feature in the prototype solution. Alternatively, it could be implemented with an enhanced narrative that represents the core feature of such mobile and web applications to unravel a distinct storyline with certain activities for individual users. The key goals of such a dominant feature would be to offer more meaning, to introduce fragments of user interactions into a common thread to achieve compelling experience that builds human connections.

These additions to prototype solution functionalities would transform unfamiliar and, for some users, complex ideas into the realm of recognisable developmental challenges each generation copes with, notably the elderly who face lack of motivation in learning new technologies, and find ICT too complex and their presentation not clear enough. Familiarizing with the interface and purpose of a digital solution through themes that are, on average, central to different developmental stages of a human being, seems a fit solution to various usability problems encountered in designing such systems. With this study, we managed to identify and prioritize the areas where additional progress can be made to maximize benefits for all involved. We will continue testing and developing the FCD approach as we assess that it should respond well to the challenges of the modern family and aging society.

### 4.3. Limitations and Future Research

There were problems that led to inconsistencies before and during the testing period. Notable problems were regarding the time period of testing and type of device users used to access the application. Families that were willing to participate in our testing of the application did not have a lot of time frame options when they were all together at home. In some cases, this restricted us to having in-person interviews and presentations only with part of a family instead of all members. There were different causes that contributed to an inconsistent time period of testing. Most of them were due to the crowded schedule of the families. Two families had problems with illness, grandmother from one family passed away during the testing period. At the beginning of the testing period, users were given tasks they had to solve and then fill out all three questionnaires. Some users were not able to use the same device for the entire testing period. The results of the questionnaires may have been affected due to this problem and should be interpreted with the type of device terminal in mind, since some of them were reported different at the end.

Triangulation approaches have certain limitations, most importantly, the potential use of different philosophies that can lead to methodological opportunism and incoherence [9]. In a practical sense, the triangulation approach can also be very time consuming. Nevertheless, these are obstacles that need to be conquered rather than fundamental objections to this approach. We deliberately used this kind of approach—triangulation with thematic analysis—in order to better review the prototype basis and fundamentals of FCD for digital services. It was used to broaden the overview of the current state rather than specifically validate the prototype in terms of requirements engineering. The execution part of our triangulation methodological approach, the thematic analysis, which is described as a flexible, discipline-agnostic method, also has to justify the demand for a clear, explicit and rigorous application. We also believe that our approach would additionally benefit if we conducted user interviews before the testing period in addition to user diaries of use between and post-usage interviews.

## Figures and Tables

**Figure 1 sensors-19-01232-f001:**
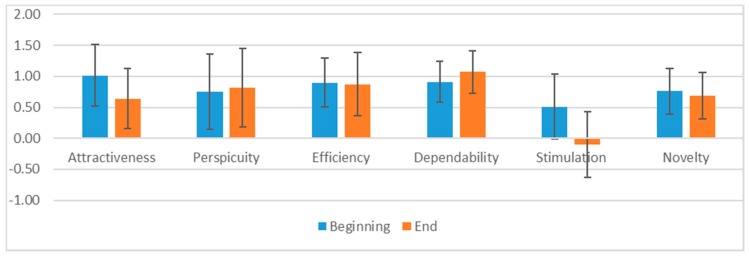
Comparison between User Experience Questionnaire (UEQ) at the beginning and at the end of user testing.

**Figure 2 sensors-19-01232-f002:**
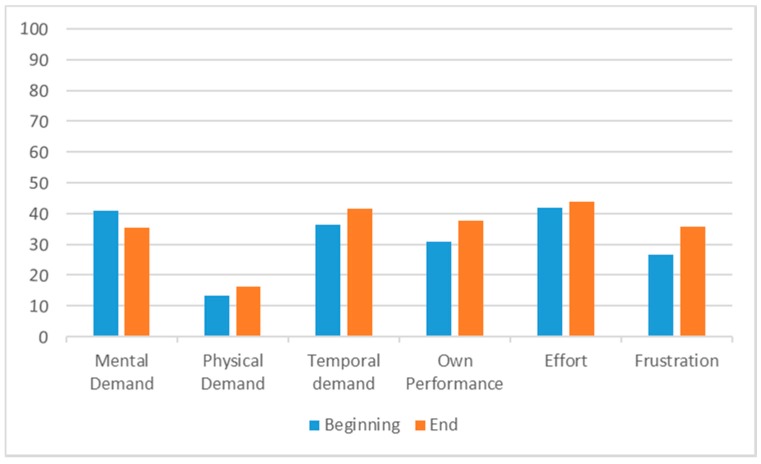
Comparison between the NASA-TLX score at the beginning and at the end of user testing.

**Table 1 sensors-19-01232-t001:** Mean and Std. dev. values for the System Usability Scale (SUS) questionnaire. This table presents mean values from data collected at the beginning and at the end of user testing, and its standard deviation.

	Beginning Mean	Beginning Std. Dev.	End Mean	End Std. Dev.
Youth (<18)	65.30	15.30	80.60	14.30
Adults	63.50	25.30	65.20	20.10
Elderly (>65)	60.80	1.4	55.80	12.60

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
