# Peer review of "Rethinking Family-Centred Design Approach Towards Creating Digital Products and Services"

_sensors, 2019, doi:10.3390/s19051232_

Reviewer 1 Report

The paper addresses an interesting topic and is well written. The motivation is described clearly, as well as the used methodology, the achieved results, and the discussion. The authors clearly point out the relevance of a multi-method and interdisciplinary approach. To summarize, the paper is self-consistent but also has a significant overlap with a previous article of the authors (Nr.2 in the list of references).  However, I see some limitations in regard to the research design and differentiation with the mentioned previous article, in terms of the selection of methodology and in reaching the goals promised in the motivation. The majority of the tools used is questionnaires, combined with interviews. Insofar, the benefits of the triangulation approach (which is well presented) is questionable. Such an approach makes more sense, in my opinion, if the variety of methods is reasonable and data from different sources is combined to consolidate results and their interpretation. In this regard the variety of the selected methods is not optimal, because they have a common drawback: they are based on retrospective and reflective assessments (in this case related the use of the "Myfamily" app). It is commonly agreed in the literature that such methods are appropriate for the evaluation of subjective aspects of interaction, but make the most sense in combination with methods evaluating more objective parameters, such as task-oriented usability tests with measures like task completion time, error rate, etc. Although the paper addresses aspects of User Experience appropriately, basic usability aspects are not covered but might influence the motivation to (frequently) use an ICT-based tool such as Myfamily significantly. Statements such as "none of the users had major difficulties in using the functionalities" (page 9) lack the empirical proof (or at least the documentation of it).  I would, therefore, propose to run an additional study (a classic usability test) covering instrumental aspects of Myfamily and contrast them to the outcome of the methods presented in the current version of the paper. This test can be (as the methods described here) qualitative, with only a small number of participants.           

Author Response

Dear reviewer, 

Thank you for reviewing the manuscript. We greatly appreciate your comments and recommendations. We have revisited the manuscript accordingly and have made changes. The following is our point-by-point responses.

Point 1:  Consider task-oriented usability tests.

Response 1: We agree on the importance of more quantitative methods. We used taks-oriented usability testing in the prototype solution design phase, which per se is not a part of the study presented in this concrete article. Nevertheless, it is very important to highlight these tests were conducted in closed laboratory environment (lines 112-128; tables in ApendixA, lines 710 trough 717) which we have done in a whole new chapter “MyFamily prototype” (lines 80 and further) for the purposes of clarity what methods were used in overall project design and implementation process made before this stage’s more qualitative insights in (more) real-life scenarios to broaden the perspective on FCD model.

 Point 2:  Revise statements in the results.

Response 2: Thank you for this input. The statement “none of the users had major difficulties in using the functionalities” was formulated to “none of the users reported…” , as this statement results from the interviews (line 465). At this particular point there is no “hard” empirical proof. The functionalities themselves were chosen and iteratively developed in accordance with the task-oriented usability testing and user experience testing to reduce the friction in the use as much as possible. This aspect was a major part of the previous studies in prototype design process, which we describe also in response to point 1.  

Reviewer 2 Report

This is a very interesting and novel research, but will be improved using a correct design of experiments as in:

Improving   an Industrial problem optimizing the material in car seats

A   Ochoa-Zezzatti, J Sánchez, A Hernández-Aguilar, R Pérez

International   Journal of Combinatorial Optimization Problems and Informatics …

1

2016

https://www.redalyc.org/html/2652/265245553007/

A correct mutivariable analysis is very important as in:

https://www.mdpi.com/1099-4300/21/1/83

In addition many features associated with Smart Cities is very important describbe when proposed digital products and services as in:

Alfred Zimmermann, Dierk Jugel, Kurt Sandkuhl, Rainer Schmidt, Christian M. Schweda, Michael Möhring:
Architectural Decision Management for Digital Transformation of Products and Services. CSIMQ 6: 31-53 (2016)

Author Response

Dear reviewer, 

Thank you for reviewing the manuscript. We greatly appreciate your comments and recommendations. We have revisited the manuscript accordingly and have made changes. The following is our point-by-point responses.

Point 1: The design of the experiment.

Response 1: In the earlier stages of the project we have conducted task-based (time to resolve a given task, critical erros,…) user experiments with standardised questionnaires in a controlled laboratory environment in order to develop usable in and easy to use features of prototype solution. We have reported results in the previous article we wrote (ICT to promote well-being within families) and we are including the summary of method and results into this article’s new chapter “MyFamily prototype” (lines 80 and further). This main focus of this article’s study corresponds to more qualitative methods in order to deepen the insight of (more) real-life scenarios to broaden the perspective on FCD model.

 Point 2: The use of multivariate analysis in methodology design.

Response 2: This article focuses on validating or omitting the selected functionalities based on post-usage interviews and questionnaires of smaller samples of users. We agree with you on the importance that pure quantitative methods in form of multivariate analysis pose. We have used statistical methods in the first half of the Ekosmart project, where we analysed broader ICT usage patterns among general population and in task-oriented usability tests in closed enviroment (lines 112-128; tables in ApendixA, lines 710 through 717).

 Point 3: The addition of Smart City features regarding digital products and services.

Response 3: Thank you for additional references regarding Smart Cities – it incorporates very well into smart city scenario planned solution in accordance with EkoSmart project solution we will eventually develop. These references provide with additional context for families in interconnected modern world (line 31; added reference, line 723).

Reviewer 3 Report

The paper presents a study based on user experience testing and interviews of a prototype solution designed for family-centered digital services.

In my opinion the paper is well written and structured, and the research method and conclusions are clearly presented. When considering the whole project (including previous publications), the used systematic approach is also very relevant.

 However, I would like to see some information (even basic) about the prototype solution and followed (or not) graphical user interface guidelines and existence or not of (pure) usability tests. Nevertheless, in my opinion, the paper can be accepted for publication.   

 By the way, it seems a little bit odd having empty sections in a document like this (e.g. 2, 2.4, 3;3.1;3.2, etc.)

Author Response

Dear reviewer, 

Thank you for reviewing the manuscript. We greatly appreciate your comments and recommendations. We have revisited the manuscript accordingly and have made changes. The following is our point-by-point responses.

Point 1: More information about the prototype solution and usability tests.

Response 1: We appreciate the comment. Whole new chapter “MyFamily prototype” (lines 80 and further) was added to the article. This was very important improvement for the article as it adds much more clarity by explaining selected prototype functionalities implemented in user interface and by highlighting task-oriented usability tests that were conducted in closed laboratory environment (lines 112-128; tables in ApendixA, lines 710 through 717) in earlier development stages.

 Point 2: Consider odd empty sections.

Response 2: Thank you for your recommendation regarding empty sections. We agree that instead of the empty sections some explanation of following chapters will be beneficial for overall style and flow of the document (lines: 174, 301, 365, 368, 442 and 565).